# Nature-Based Solutions and Real-Time Control: Challenges and Opportunities

**José Brasil** [1,*], **Marina Macedo** [1], **César Lago** [1], **Thalita Oliveira** [1], **Marcus Júnior** [2], **Tassiana Oliveira** [1] **and Eduardo Mendiondo** [1]

1   Department of Hydraulic and Sanitation, EESC—University of São Paulo, São Carlos 13566-590, Brazil; marinabatalini@usp.br (M.M.); cesarlago@usp.br (C.L.); thalitaoliveira@usp.br (T.O.); tassiana.halmenschlager@usp.br (T.O.); emm@sc.usp.br (E.M.)
2   Department of Civil and Environmental Engineering, University of Texas at San Antonio, One UTSA Circle, San Antonio, TX 78249, USA; marcus.gomesjunior@utsa.edu
*   Correspondence: arturtbr@usp.br

**Abstract:** Nature-based solutions (NBS) as green infrastructures to urban drainage are an effective mitigation strategy both in terms of quantity and quality of runoff. Real-time control (RTC) can complement both flood mitigation and improvement of water quality by controlling elements of the drainage and sewage system. This study assessed the improvement opportunities with RTC of three NBS-related techniques commonly applied in urban drainage with different spatial scales: green roof, bioretention and detention basin and the remaining challenges to integrate both methods. Additionally, our investigations showed that the main difficulties reported involve the planning and monitoring stages of the RTC system. All of the studied devices can benefit from RTC. It is possible to observe that, despite the good results reported in the literature, the application of RTC to NBS studies on urban drainage are very recent. There are several opportunities that can be explored to optimize the performance.

**Keywords:** nature-based solutions; real-time control; urban resilience; water quality improvement

## 1. Introduction

Urban development and impervious surface growth increase the volume of stormwater runoff, peak flow, pollutant loads and concentrations, in addition to reducing the runoff time of concentration [1]. Moreover, climate changes tend to aggravate the impact on urban drainage systems due to the increase in the occurrence of extreme events [2]. The traditional solutions to flood control are to enlarge the drainage system infrastructure or expand the capacity of existing structures to rapidly transport water downstream [3,4]. However, the costs associated with these solutions are high and, as urbanization advances, new expansions are needed at the systems.

Therefore, stormwater source control approaches have gained prominence as viable solutions for improving urban resilience, which, in a decentralized manner, allow a reduction in the impact of natural disasters and impacts caused by environmental changes both in terms of runoff volume and water quality [5–7]. Nature-based solutions (NBS) are defined as living solutions in which processes and structures are designed to meet different environmental challenges while simultaneously providing several economic, social and ecological benefits [8,9].

Regarding urban drainage, NBS are associated with the concept of Low Impact Development (LID). In a historical context, LID initially aimed to reintegrate excess runoff into the hydrological cycle [10]. With the advances of techniques, LID incorporated objectives of mitigating the impacts caused by climate change, in addition to proposing to reintegrate the runoff volume into the catchment through local reuse [11,12]. Therefore, incorporating medium- and long-term time scales, considering the increased occurrence

of extreme flood and drought events and aiming at the recycling and co-management of resources, LIDs can contribute to several United Nations Sustainable Development Goals (UN SDG), specifically, the objectives of zero hunger (SDG 2), clean water and sanitation (SDG 6), affordable and clean energy (SDG 7), sustainable cities and communities (SDG 11) and climate action (SDG 13).

LID practices provide reductions in peak flow and diffuse pollution through physical–chemical processes, such as filtration and drainage layers at the systems [10]. However, despite the most varied designs, these systems are projected to function passively, i.e., they do not adapt the system configurations over time to optimize aerobic, anaerobic and/or hydraulic processes at the layers [12].

Therefore, the application of real-time control (RTC) can assist in the system operation and optimization. The RTC allows the modification of the state of one or more elements by adjusting the system configuration based on continuous monitoring and/or forecast [13]. The first RTC application in urban drainage was implemented in Minneapolis (USA) in the late 1960s [14]. Since then, RTC strategies have been developed and applied in several urban drainage systems around the world, especially in Europe and North America, generally with the objectives of reducing storage tank volume in Combined Sewage Overflow (CSO) systems to prevent urban flooding or minimizing operational costs [15,16]. Centralized control strategies request a control room, which receives the monitoring data and operates the actuators with coordination. That way, this strategy is recommended when all the actuators should operate jointly or when a global system control is necessary [15]. The distributed control strategies use several components of systems, also called nodes, working at the local level, allowing global system simplifications [17]. More sophisticated systems that rely on automatic sensors, acquisition, data processing and control of the actuators can be expensive to implement [18]. The sophistication level of RTC depends on the objectives of the management plans and local site characteristics. Therefore, it is important to understand the advantages of each NBS type and how RTC enhances their performance according to their specificities. This knowledge can help decision makers to elaborate a more accurate and cost-efficient system of NBS with RTC.

Some studies explored RTC application in urban drainage systems combined with passively functioning NBS [19,20]. However, recently, new studies have been carried out to evaluate real-time controlled NBS to promote adaptability and optimize hydraulic/hydrologic and water quality objectives of these techniques. Therefore, this paper aims to explore the opportunities for joint application of both techniques (NBS + RTC), to report the main challenges found in the literature regarding the integration of RTC and to perform a comparative analysis of NBS techniques and their potential benefits associated with the application of RTC. The identification of how RTC can contribute to different techniques, improve the benefits and identify the application challenges aims to contribute to the 16th unsolved problem in hydrology: "How can we use innovative technologies to measure surface and subsurface properties, states and fluxes at a range of spatial and temporal scales?" [21].

## 2. Opportunities of RTC in Different NBS

We selected three different techniques to analyze different application scales. At the property scale, green roofs were selected. At the street scale drainage system, bioretention was selected. Finally, at the neighborhood to watershed scale, detention basins were investigated. A brief literature review on the operation, design and experimental results of these techniques was carried out, as well as, when applicable, the history of RTC application. For each technique, it was also indicated where the RTC can act to enhance the existing benefits or even bring new uses to the device.

### 2.1. Green Roof

In developed cities, roof areas account for about 40–50% of total impermeable urban surfaces. Therefore, in urban areas, the application of green roofs has growing attention due to environmental benefits and energy efficiency. A green roof is a layered system

with a waterproofing membrane, growing media (usually a soil mix) and the vegetation layer. Green roofs often include a root barrier layer, drainage layer and, where the climate requires, an irrigation system [22].

Green roofs can also store water for re-use purposes, which offers a sustainable and aesthetical treatment solution to its irrigation at dry periods or as a supplemental water to non-potable use [23]. To increase storage capacity on green roofs, aiming at both reuse and runoff retention, an adaptation called green-blue roof was proposed. The green blue roof is a modified green roof with an additional storage layer to reduce peak flow, promote water reuse and reduce heat waves at the roof surface [24]. A green roof structure without internal storage and with internal storage working as a green–blue roof is shown in Figure 1.

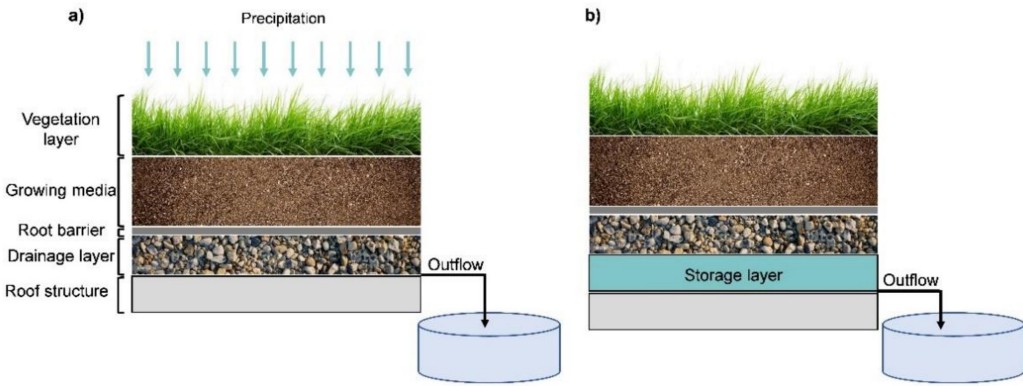

**Figure 1.** Green roofs designs. (**a**) Usual layers of a green roof. (**b**) Example of a green–blue roof with storage layer to prevent flash floods. In both cases, the use of a stormwater harvesting reservoir connected with the outflow is optional and not so common; however, it adds more possibilities for real-time control (RTC).

No specific literature was found on RTC applied directly to green roofs. However, there are several opportunities to optimize green roof behavior when operated in real-time. The success of green roofs depends on plant health both for improving water and air quality and reducing heat waves [21–24]; however, long dry periods can affect the health of plants if they are not properly maintained. Through the addition of a stormwater harvesting reservoir in the system, it is possible to increase the versatility of both devices. Reusing water from conventional stormwater harvesting systems can be limited by water quality factors [25,26]. In addition, conventional stormwater harvesting systems have frequent overflows, as the reservoir remains almost full due to lack of water demand for reuse [27]. In this way, the water stored in the stormwater harvesting system can be used to maintain the green roof through the monitoring of soil moisture and an RTC system, while maintaining optimum points of plant growth, as the green roof can treat the water and increase the possibilities of reuse, as for non-potable demands. The addition of stormwater harvesting reservoirs to green roofs can also benefit from reduced runoff and peak flow, especially when used in RTC [28–30]

Studies show that green roofs can retain stormwater to reduce peak flow and runoff [31–33]. In [20], the application of movable gates with RTC was studied in an urban basin to reduce CSO by comparing scenarios using only RTC and RTC combined with passive green roofs. In [34], a history of studies related to RTC at the same basin was presented, and the best performance obtained for reducing overflows was with the combined configuration of RTC + passive green roof. In [19], similar results were achieved when associating different configurations of real-time controlled sluice gates with scenarios containing green infrastructures in the system, such as the green roof, permeable pavement and stormwater harvesting reservoir. These combined effects of RTC application in drainage systems along with other mitigation techniques, such as NBS, are an important evolution of RTC applied to urban drainage; however, the benefits can be enhanced with actively real-time controlled NBS. In addition, green roofs operated by RTC can store water for longer periods by closing

a control valve, enhancing the heat reduction effect, and creating anaerobic zones for water treatment [35,36].

### 2.2. Bioretention

Bioretention systems are techniques that use soil compositions and vegetations to promote stormwater runoff infiltration, evapotranspiration, pollutant removal and uptake by plants, in addition to providing aesthetic and social benefits [1,37]. Generally, bioretention systems are composed, from top to bottom, of a ponding zone; a vegetated layer; a soil mix media layer, also called filter media; and a drainage layer typically composed by coarse aggregate.

The literature shows favorable results both for runoff control and water quality improvement. Numerous studies have indicated that bioretention systems are useful to control runoff in urban areas, with runoff reduction of 36–96%, depending on the rainfall intensity [1,38,39]. When the inflow arrives at a bioretention cell, the runoff infiltrates the system layers, filling the soil pores until it reaches the drainage layer. This water retention affects the efficiency of the runoff reduction, where the previous soil moisture is the main factor impacting the capability to retain water [1]. In the drainage layer, runoff can continue to infiltrate the soil, supply the underground aquifer (exfiltration) or, in the case of bottom lined bioretention, an underdrain can direct the water to a stormwater harvesting reservoir, preferably, or back to the drainage system.

Despite these results, some design alternatives can significantly affect the bioretention performance of a given parameter with passive operation. Regarding the drainage layer design, free drainage (FD) bioretention has shown good results in the literature but is dominated by aerobic conditions and lacks in performing both nitrification and denitrification [40–42]. Therefore, bioretention cells with internal water storage (IWS) promote aerobic and anaerobic conditions favoring both nitrification and denitrification [13,43], although compromising the bioretention hydraulic performance [35,44]. A typical bioretention structure, for FD and IWS design, is shown in Figure 2.

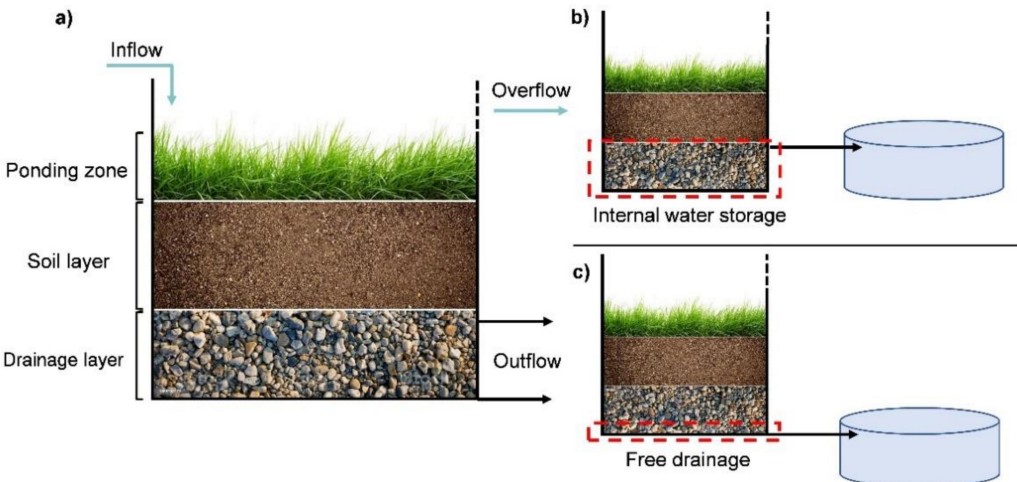

**Figure 2.** Standard bioretention designs. (**a**) Typical bioretention structure with three layers. At (**b**) a stormwater harvesting reservoir is added at a height that allows the creation of an internal water storage (IWS). (**c**) represents a free drainage (FD) bioretention. In both cases, the stormwater harvesting reservoir is optional; however, it adds more possibilities for RTC.

In [12] several RTC schemes in bioretention columns to improve water quality were analyzed. The most notable water quality improvement was related to nutrients, while the static system performed best for ammonium by more than 40%, and columns with internal water storage removed nitrate by more than 73%. They also suggest that active control and RTC may strike a balance between traditional free drainage and internal water storage systems. However, the study did not access the trade-off between water quality and

hydraulic performance. In addition, the maximum dry period during the study was 9 days, in which it was not possible to evaluate the dry/wet cycles. These cycles are important to microbial activity and, consequently, metal and nutrient removal, and affect the plant health; furthermore, long dry periods cause an increase in pollutant build up and their consequent wash off in the next rainfall event [45,46]. In this way, the dry/wet cycles can be managed by maintaining the IWS zone with RTC even in bioretention cells that work with FD by the outflow control through an automated valve [26].

Another important decision in the design of bioretention cells is related to the outflow water storage for reuse purposes. For this, it is necessary that bioretention provides an efficient treatment of both pollutants and pathogens to comply with specific regional legislations for reuse [47–49]. Bioretention cells controlled in real time can benefit in several ways, either through the treatment improvement stimulated by the creation of IWS, through the by-pass of first flush for the high load of pollutants, or, in the case of water quality real-time monitoring, the by-pass of the water entering the reservoir back to the drainage system if the maximum limit of the respective legislation is met. In [49], the treatment of stormwater for harvesting and reuse through two low-cost implementation strategies for bioretention cells was studied. The authors found that bioretention with RTC can mitigate the negative effects of short and long dry periods and mitigate the influence that large inflow volumes have on the treatment of fecal microbes. Compared to passive bioretention cells, RTC configurations significantly reduced the amount of total suspended solids (TSS), total phosphorus (TP) and total nitrogen (TN), although the study did not analyze metals and the nitrogen parcels corresponding to nitrite and nitrate, which can be considered limiting for reuse purposes [47]. In addition, the authors did not assess the impact of inaccurate forecasts on the performance of bioretention with RTC. Furthermore, further studies are needed to assess the viability of bioretention RTC with rain forecasting models. Bioretention cells have the potential to improve water quality. However, there are different quality standard limits for different reuse types, and additional treatment may be required. In [47], the water quality of a bioretention cell outflow was analyzed, and it was observed that for USA legislation for water reuse, only turbidity was inadequate, while for Australian legislation, turbidity, $NH_3$, Fe, Pb, Ni, Cd and color were outside the normative standard. Increasing its performance with RTC, as these studies showed, can enhance the potential for water reuse.

Scientific gaps still exist for RTC in bioretention systems, such as questions about the centralized analysis of several bioretention cells, quantitative aspects of runoff and peak flow reduction and multiple RTC objectives.

*2.3. Detention Basin*

A detention basin temporarily stores a portion of the incoming water volume in selected areas to manage floods [50]. Its outflow depends on the type and size of the outlet structure. A detention basin has at least one primary outlet, such as an orifice, weir or riser type outlet, to pass the regulated flow from the basin and one secondary outlet (emergency weir) to pass the overflow above the basin storage during rainy seasons [51]. Conventional detention basins are designed for flooding protection purpose. This measure has little water quality benefits, although some design strategies can improve the pollutant removal. In [52], a two-stage detention basin was developed. The bottom stage has a small outlet to promote settling of pollutants, while the top stage remains dry, except during large storms. The detention basins can also be designed to be kept dry between storm events (dry detention basins). These have the advantage of preventing mosquito proliferation on standing waters [53]. This is also a concern that needs to be considered while planning both traditional and sustainable drainage systems [54,55].

Detention basins can also be classified as in-line or off-line (Figure 3). While in-line basins may cause water to build up and flood land upstream, off-line basins do not create any barriers to sediment transport and fish migration [50]. In-line detention basins operate from a wide range of inflow [56], which is dependent on weather conditions. On the

other hand, off-line detention basins use diverged flow, which can be controlled and is more predictable, although the reduction in runoff volume is smaller compared to in-line detention basins. The control of the inflow can be an additional operation during RTC of the off-line configuration, which is usually performed at the outlet. The flow diversion starts after the river flow reaches a threshold. Therefore, the selection of the proper threshold value is essential to an effective flood control [50]. RTC can make use of the state variables of the system to decide whether to start the diversion and optimize the performance of the drainage system. In [57], for instance, the active control of flow diversion was applied to an off-line detention basin, and the flood reduction was compared with the passive control of the structure. Their results showed that flood reduction with the active control was 2–3 times higher than the passive one.

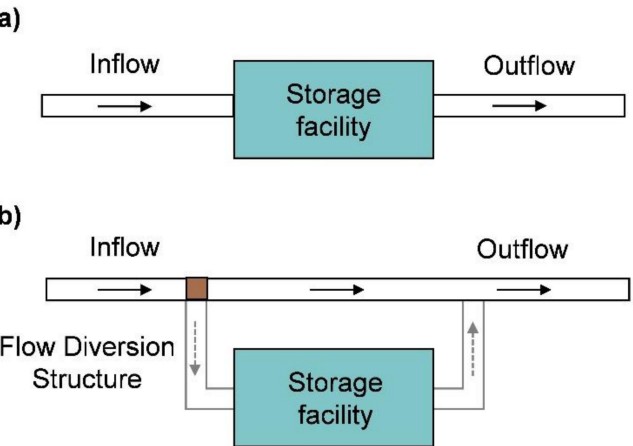

**Figure 3.** Examples of in-line structure of a detention basin (**a**) and an off-line detention basin (**b**).

Among the three NBS analyzed, detention basins are the most studied in RTC applications. Historically, the firsts RTC assessments in detention basins were experimental through on/off valve control or orifice openings to increase the detention time and consequently increase the TSS and pathogen removal for retrofit purpose [58–60]. However, detention basin control structures were incipient and had basic operation rules (e.g., outflow release with reservoir level measurements at 20, 30 or 50%), and then, the maximum storage, the detention time and pollutant removal at the detention basin were evaluated. Recently, in [61], the experimental control of the downstream flow was attempted through integrated parallel detention basins with modular valve openings. The study presents simple but effective RTC configurations for flood control, as it works in an integrated way with global communication of the agents in the drainage system.

In parallel to the experimental studies, several other studies were developed with a modeling approach to both the drainage system and the detention basin behavior. In [18], an important study was developed in which new operational rules were added to detention basins control. The authors sought to optimize the detention time to avoid overflows, reduce the first flush effect, slowly discharge the outflow to minimize the hydraulic shocks of the receiving bodies and minimize the solids resuspension and limit the detention time to prevent mosquito breeding, in addition to minimizing the gate operations to prevent the rapid wear and tear of the actuator. These operational rules illustrate the RTC potential gain in detention basins, since it is possible to meet multiple hydrological/hydraulic and water quality objectives, in addition to optimizing the device operation. Moreover, in [62,63], the authors sought to assess a reduction in overflows and increase detention times between predictive and reactive RTC systems. The results showed that predictive systems, even influenced by errors, have better performances than reactive systems, which incorporates into RTC applications the possibility of anticipating the system and preparing within a sufficient prediction horizon to optimize operations. Furthermore, in [64,65], studies were

developed focused on the water quality improvement of detention basins, obtaining good results for the TSS and metal and nutrient removal for different RTC configurations. Even with strategies focused on water quality, it was possible to observe an improvement in the hydraulic performance of the detention basins studied.

Multiobjective RTC could be applied to maximize TSS reduction, and therefore, reduce contaminants while minimizing floods downstream. Additionally, there is a lack of application to RTC and optimization methods to universally integrated systems for stormwater structures at the operational level that perform optimally under varying environmental conditions [56,66]. In [67], a framework was developed and RTC was applied in four detention basins integrated to the urban drainage system of an urban basin in Canada. As a result, it was possible to reduce peak flow, increase detention time and improve control on overflows for both the observed and climate change scenarios. In addition, there was a significant improvement in the flow velocity when the detention basins were operated with RTC, by reducing erosion in the basin. Another important contribution was the analysis of the impact of rain forecast errors, since when errors are considered, the performance of the device drops (although it is still better than with static rules), and there may be overflows occurrences.

## 3. Challenges to Applying RTC to NBS

Planning an RTC system is complex [15]. Some decisions must be made in the planning stage, as it is a system that has a high implementation cost. An important decision at this stage is to define whether the system will be centralized or distributed. Centralized systems tend to perform better, since the management is performed in an integrated way; however they are more complex, expensive and require central control [16]. Due to the complexity of the urban drainage systems, where changing a part of it can change the behavior of the entire system, centralized RTC applications are recommended. There are several studies of applications of centralized RTC systems in urban drainage systems [68–70]; however, most RTC applications are carried out with the purpose of reducing the combined sewer overflow, while having little regard for RTC applications in separated sewer systems [66]. The applications of distributed RTC have economic advantages, considering the local level optimization, not including the complexity of a drainage system. In [71], the hydraulic performance of an urban drainage system was evaluated using decentralized RTC techniques based on gossip algorithms, which reduced the complexity compared to centralized systems, although improving the system's hydraulic performance.

Alternatives aiming at spatial distribution, such as green roofs, include even greater complexity compared to centralized systems, since they would require more equipment for monitoring and control; it is more feasible in these cases to use a more decentralized approach. Regarding bioretention cells, the scale of application can influence the decision on the type of control. As for detention basins, the centralized approach is more viable, as it can adapt its operation depending on the state of the drainage system and is a device usually located in low points of the basin.

### 3.1. Monitoring

Another important factor in the implementation of RTC systems is the monitoring and control equipment. There are several specific manuals for monitoring each technique for NBS applied to urban drainage [72,73]. For infiltration devices that offer some type of water treatment, such as bioretention and green roof, parameters related to water quality and the system water balance must be monitored. The water balance variables to be monitored for the infiltration techniques can be summarized as precipitation, soil moisture, inflow, outflow, soil infiltration rate and, when applicable, stored water level. As for the water quality monitoring, the parameters commonly evaluated are nutrients (phosphorus and nitrogen), metals and total solids (TS). Meanwhile, for devices such as detention basins, which act as a reservoir for flooding amortizing, the monitoring of water balance variables is limited to the inflow and outflow, reservoir level and, in some cases, evaporation, while

the monitoring of water quality is on analysis of sediments, as this is usually the only type of pollutant treatment present in detention basins.

However, the manuals are limited to the specific monitoring of NBS at the site level. In addition, for the application of the RTC, it is necessary to have integrated and real-time monitoring [74]. Monitoring all parameters related to NBS may be unfeasible as the number of devices grows (e.g., monitoring many green roofs can be time and economically consuming); therefore, devices with a larger spatial scale have more advantages in monitoring. Thus, for smaller spatial scale NBS (e.g., bioretention and green roof), simplifying the number of monitored parameters may be necessary for the application of the RTC. Table 1 shows which parameters are most commonly monitored for the different types of NBS and the impacts of monitoring these parameters on the techniques evaluated considering an RTC scenario.

**Table 1.** Monitoring impacts parameters on the evaluated techniques considering an RTC scenario.

| NBS | Precipitation | Inflow | Outflow | Soil Moisture | Infiltration | Et | Reservoir Level | Nutrients | Metals | TS |
|---|---|---|---|---|---|---|---|---|---|---|
| GR | High | Low | High | High | Low | Low | Low-Medium | High | High | High |
| BT | High | High | High | High | Medium | Low | Low-Medium | High | High | High |
| DB | High | High | High | - | - | Medium | High | Low | Low | High |

The main parameters of the evaluated techniques were considered. The ratings represented by "-" mean that monitoring does not apply to the technique in its usual design. NBS: Nature-based Solutions; GR: green roof; BT: bioretention; DB: detention basins; ET: evapotranspiration.

The green roofs do not have water inlet structures due to their spatial distribution characteristic; therefore, monitoring precipitation is sufficient to characterize the inflow. In the case of bioretention with RTC, monitoring the inflow is essential to assess the input volume so that it is possible to carry out the optimal controls of the system to achieve the objectives, whether they are related to water quality or reduction in the runoff volume and peak. For the application of RTC in both techniques, the outflow monitoring also has a high impact, since if the devices have a reservoir for reuse, the outflow of the device would function as the inflow of the reservoir. In addition, with the joint monitoring of the inflow and outflow, it is possible to determine the volume of water that remains inside the device for a certain period.

For the detention basins, evaluating the inflow, storage level and outflow is essential to ensure that overflow does not occur, since an overflow can bring losses and floods to both the perimeters immediately around and downstream of the device due to the spatial scale of the device. The evaporation of water can be significant in specific cases (e.g., detention basins with large mirrors of water and/or in tropical/sub-tropical areas).

Both the runoff retention and peak flow attenuation efficiency of bioretention and green roofs are linked to the soil mix infiltration capacity. Green roofs are installed in impermeable covers, just as the soil and vegetation layers are thin compared to other techniques to avoid adding weight to the roof [75,76]. Therefore, the impact of monitoring infiltration was considered low. In bioretention, the impact was considered medium, because the variation in the infiltration rate occurs mainly due to the clogging effect that occurs slowly and varies depending on the soil and vegetation, despite being a fundamental parameter for the correct functioning of the device [39]. Consequently, for this parameter, monitoring can be carried out sporadically in bioretention.

Monitoring soil moisture has a high impact on both bioretention and green roofs. Dry/wet cycles are essential components in these devices, since, in dry periods, the biological processes responsible for the metal treatment and microbial dormancy can affect the pollutant removal performance of the devices, in addition to causing mineralization and accumulation of organic matter due to inactivity of the microorganisms responsible for the treatment [12,46,77]. With RTC, devices can be more dynamic, controlling the IWS on the device and consequently controlling the dry/wet cycles. Previous humidity conditions

also affect the water balance performance of the devices [43] and the metal absorption by hyper-accumulating plants [78].

Among the parameters that require monitoring, the water quality-related parameters are the most complex. Traditionally, water quality monitoring depends on sampling and laboratorial analysis, which is an expensive, laborious process, leading to limited spatial and temporal scales [79]. However, real-time monitoring of water quality for RTC purposes is becoming more recurrent with the advance of technologies applied to sensors and the development of the Internet of Things (IoT) in the last decade [80,81].

For the detention basins, it is important to monitor total solids. The pollutant reduction efficiency of the detention basin is a function of several variables, which include the physical, chemical and biological characteristics of the pollutants, as well as the precipitation regime, detention time and geometric characteristics [82]. However, it is possible to simplify the water quality monitoring of the detention basins by monitoring the detention time, removing suspended solids with the pollutants associated with them and allowing disinfection by UV light during the day [18,83,84]. In [67], for example, detention time was used as one water quality-related parameter to be optimized in an RTC application in detention basins. Although there are reports of improvements in water quality regarding nutrients and metals [84], the treatments promoted by detention basins for these parameters are not as effective as the treatments offered by green roofs and bioretention.

However, the NBS monitoring improvement with RTC application also enhances other components of the urban drainage system. Locations vulnerable to flood events can benefit from the enhanced qualities of NBS with RTC as discussed previously, but they can also benefit from more weather data from real-time monitoring used to flood early warning systems. The Center for Studies and Research in Disasters of Federal University of Santa Catarina—CEPED UFSC, for example, reported 9002 natural disasters generating losses of BRL 72 billion (approximately USD 13 billion) between 1995 and 2014 [85]. In [86], through flood risk management actions scenarios, the difference in cost per inhabitant per urban area was assessed. Scenarios that include monitoring and alert systems cost approximately 11 USD.inhab$^{-1}$.m$^{-2}$, while systems that do not have monitoring and alert systems based on reconstruction after natural disasters cost an average of 19 USD.inhab$^{-1}$.m$^{-2}$. In addition, NBS can be installed primarily in vulnerable locations, with multiscale coverage to integrate stakeholders and associated costs [87,88]. Thus, further studies with real-time controlled NBS must be conducted in order to assess the economic and social benefits that hydrological, water quality and monitoring can generate for urban environments.

### 3.2. Modeling, Forecast and RTC Strategies

There are several RTC strategies that can be adopted in urban drainage systems. In [15], the main RTC techniques used in the literature were reviewed and divided into heuristic and optimization-based algorithms: the heuristics were the rule-based control and fuzzy logic control, while the optimization-based ones were the linear-quadratic regulator, evolutionary strategies, model predictive control and population dynamics-basic control. Although heuristic approaches have low complexity, these algorithms do not guarantee the optimal solution and depend on experience and knowledge of the system [69]. Optimization-based algorithms, on the other hand, involve an optimization problem that represents the desired behaviour of the system. Thus, the objectives can be simultaneously or not related to problems of water quality, runoff quantity or costs related to urban drainage [89]. However, most of these algorithms require an understanding of the dynamics of the drainage system in an integrated way so that it can then perform the optimal control actions [90,91]. Therefore, depending on the RTC algorithm to be implemented for the control of NBS, it is necessary to model the urban drainage system related to the runoff and/or water quality parameters.

Empirical, black-box, process-based and ultimately physical-based models have been extensively applied to model hydrological process in different watershed scales [92–96]. Their spatial application is typically lumped, semidistributed or fully distributed or dis-

cretized. Fundamental hydrological equations are solved to estimate outputs (e.g., overland flow, groundwater recharge and evapotranspiration) in each individual hydrological unit (e.g., cell in a grid or delineated watershed). On one hand, black box/statistical models can rapidly provide good results when relatively extensive observations are available for calibration. On the other hand, they typically fail to explain outputs for inputs outside of its observation range and are not based on the problem physics. Physical-based or process-based approaches, however, can explain local details and the boundary conditions that are not typically explained in distributed empirical models, even when applied in a suitable resolution. These details can be important, especially for modeling sparce NBS infrastructure effectiveness [97]. Nonetheless, they require several different parameters to describe processes such as: (i) infiltration, (ii) rainfall spatial–temporal distribution, (iii) evaporation/evapotranspiration and (iv) pollutant transport and fate. The most suitable alternative is a function of (a) data acquisition/availability, (b) watershed scale, (c) computational and memory resources and (d) modeling time.

Flood routing, overland flow and pollutant transport and fate are directly connected in urban environments as highly impervious cities. Modeling pollutant concentrations and loads for high-frequency and high-resolution time and space requires the spatial estimation of pollutant accumulation (build-up) and pollutant washing (wash-off). This approach, however, has been applied to roughly estimate pollutant accumulation as a function of antecedent dry days and pollutant transport in terms of exponential or power fitness functions calibrated with observations [98,99]. However, these parameters are site sensitive, especially because local conditions, such as wind direction and speed, road characteristics, land-use and land-cover, can play a vital role in the spatial variation of pollutant accumulation [100]. In [101], the use of total mean daily loads (TMDL) or event mean concentration (EMC) as an alternative to high-resolution pollutographs is recommended due to the high variance of pollutant conditions.

The application of real time-control strategies to model stormwater quantity and quality needs to manage decision time and accuracy. A promising scenario would be a fully distributed physical-based watershed model, coupled with individual models for each decentralized NBS practice, interacting with the watershed. The control of NBS practices can be performed by a quadratic optimization problem, minimizing a cost function that can capture not only flood risks but also water quality reference levels, as well as not allowing large variations in the valves for a small time. This type of approach applied for model predictive control has been applied to autonomous cars, water network systems and can be applied to manage water resource references.

Rain forecasts and the current hydrological status of the system can determine the necessary controls for the drainage infrastructure to ensure adaptability [102]. While the hydrological status of the system can be obtained through the monitoring network necessary for RTC application, rain forecasts can be made through meteorological modelling. However, the rain forecasting models have several uncertainties, such as of the model itself, of the parameters used and of the precipitation non-stationarity due to climate changes [103,104].

Regarding this issue, in [105] the Dynamic Over-flow Risk Assessment (DORA) for RTC in urban drainage systems was developed, where the uncertainties of the rain forecast and, consequently, the uncertainties on the runoff are evaluated to decrease the risk of overflow in cases of incorrect forecasts. Although the DORA analysis was performed for CSO, the assessment of the uncertainties related to the forecast should also be integrated into RTC systems with NBS. In [67], for example, the impact of errors on the outflow of four detention basins installed in an urban basin applied with the RTC was evaluated. The authors compared perfect predictions from observed data with predictions made by the High Resolution Deterministic Prediction System (HRDPS), and as a result, they observed that using the prediction through the deterministic model instead of the perfect prediction led to at least one overflow in each detention basin during the simulated period. Despite

the overflows caused by the uncertainty in the forecast, the dynamic performance of the system allowed a fast and reliable recovery.

However, the computational burden becomes a limitation of the application of these systems when using forecasts (especially when using long forecast horizons), since the solutions must be computed in real time between the control actions [106]. An alternative to reduce the forecast horizon is to use values close to the catchment time of concentration. In [107] it was found that when optimizing urban reservoirs with RTC in an urban catchment with a time of concentration of 1 h, increasing the forecast horizon to 2 h obtained better results than those obtained with a forecast horizon of 1 h. However, there was no significant improvement by increasing the forecast horizon to 3 h. This happened because the forecasting capacity of the system decreases as the forecast horizon increases. Due to the importance of the forecast horizon for the RTC, future studies should be conducted with the objective of determining the impact of the choice of forecast horizons on the urban drainage systems, as well as assessing the evaluation methodology based on the forecast model used.

The reliability of the system must be discussed, since a failure could impact both the runoff and water quality, causing flood events and decreasing water quality. For example, the flow diversion to an off-line detention basin needs to be activated before peak flow reaches the structure; otherwise, the active control can have a worse performance than passive control and increase flooding [57]. This case highlights how forecasts can improve RTC. However, the uncertainties can hamper RTC and reduce its performance. Consequently, the maintenance of the system cannot be neglected due to the negative effects that can be caused by the system failure [18,108]. Although little is mentioned in the RTC literature on urban drainage systems, it is also important to account for cyber security. Since RTC systems rely on online connections to receive data and send controls, hacker attacks can impair the functioning of the devices.

## 4. Comparative Analysis and Discussion

Despite the differences in the scales of the analyzed techniques, it is possible to make a comparison between the different NBS to summarize the discussion on the opportunities and challenges of RTC implementation. Table 2 summarizes the comparative analysis.

**Table 2.** Comparative analysis on discussed NBS to RTC implementation.

| NBS | Literature on RTC | Distributed RTC Possibilities | Centralized RTC Possibilities | Implementation Costs [1] |
|---|---|---|---|---|
| Green roof | − | + | −+ | ++ |
| Bioretention | + | ++ | + | −+ |
| Detention basin | + | + | ++ | − |

[1] The implementation costs depend on the sensors and equipment that are usually found in the respective literature of each technique.

From the studies on the application of RTC in NBS, it is possible to observe that few papers have been published on the topic. Since no specific studies for RTC were found on green roofs, this technique was evaluated as having the lowest performance.

For distributed RTC applications, bioretention presented a greater potential than other techniques, since bioretention cells allow good results, both in terms of runoff quantity control and water quality improvement. However, there is also a potential for improvement for other techniques in a decentralized application form. We emphasized green roofs, as they are applied on a smaller scale and require a reservoir structure that is not always present in its design.

For centralized applications, detention basins are more highly recommended, as they work on a larger scale and can be integrated with the drainage system. Green roofs can also benefit from centralized controls; however, the complexity is relatively high, since for a watershed drainage scale it would be necessary to apply many green roofs for a

significant effect on the urban runoff. As the literature on bioretention shows applications with different scales, the device can benefit from centralized controls for specific cases of application in watershed scales.

The scale of the green roofs also helps to increase cost, since each green roof must have its own monitoring structure, leading to a higher number of sensors employed and a sending and receiving data structure that is more expensive than for the other techniques. For bioretention techniques, the cost of implementation depends on the device design and the monitoring strategies. For detention basins, most devices are already installed or in operation, making real-time control cheaper than when compared to other techniques.

Monitoring NBS, especially those with a smaller scale, can derail the implementation of RTC due to the large number of variables whose monitoring is necessary. Future investigations focusing on which parameters can be monitored indirectly or obtained through models with simpler monitoring parameters could be a significant advance for the study of RTC in NBS. Table 3 shows a summary of the RTC in NBS studies.

**Table 3.** Synthesis of studies which uses RTC on NBS.

| Authors | NBS | Analysis | Approach | Performance Criteria | Rainfall Forecasts |
|---------|-----|----------|----------|----------------------|--------------------|
| [58] | DB | Water quality and hydraulic performance | EXP | Hydraulic interception rate, pollution removal rate and maximum storage capacity | - |
| [59] | DB | Water quality | EXP | Cu, Pb Zn (total and dissolved), COD, TN, $NO_3^-$, $NO_2^-$, TKN, P (total and dissolved) and TSS | - |
| [18] | DB | Water quality and hydraulic performance | MD | TSS removal, time at several Q thresholds, overflows and time excess | Perfect forecast and error-containing forecast |
| [65] | DB | Water quality | MD | TSS removal | - |
| [64] | DB | Water quality | MD | Zn, Mn, TSS and $NH_3$-N removal | Error-containing forecast |
| [60] | DB | Water quality | EXP | *E. coli*, $NO_3^-$-N, $NO_2^-$-N, TKN, P (total and dissolved) and TSS | - |
| [62] | DB | Water quality and hydraulic performance | MD | TSS removal, time at several Q thresholds, overflows and time excess | Several forecast sources and perfect forecasts |
| [61] | DB | Hydraulic performance | EXP | Change in flow due successive valve activations | - |
| [63] | DB | Water quality and hydraulic performance | MD | Water depth, peak flows, detention time and percentage of cross-section area filled by water in the collector pipe | Perfect forecast |
| [12] | BR | Water quality | EXP | Cu, Zn, Mn, $NH_4^+$-N, $NO_2^-$-N, $NO_3^-$-N and TSS removal. | Error-containing forecast |
| [49] | BR | Water quality and hydraulic performance | EXP | Stormwater: volume harvested, discharged to environment, evapotranspired, load removal (TSS, TN, TP), load harvested and load discharged to environment. | - |
| [56] | DB | Water quality and hydraulic performance | MD | Duration curves, water level and TSS | - |
| [66] | DB | Water quality and hydraulic performance | MD | Peak discharge, quality improvement, overflow prevention, improved flow attenuation and outflow variation minimization | Perfect forecast |
| [67] | DB | Water quality, hydraulic performance and erosion | MD | Peak discharge, quality control, overflow control, mean flow variation and flow velocity | Perfect forecast and error-containing forecast |

DB: detention basins; BR: bioretention; MD: modeling; EXP: experimental.

It is possible to observe that detention basins are, in fact, the NBS with the largest number of studies and variability of analyzed parameters. Using this technique, experimental and modeling studies have been developed with RTC application in multiple detention basins integrated with the urban drainage system. As for rainfall forecasts, the studies explored both those applied with perfect forecasts and those with errors. Despite the number of studies, there is a lack of different climates, regions and topographies that can be explored in further analysis. Table 3 also shows that studies related to real-time controlled bioretention are recent. In addition to experimental analysis, modeling approaches are a potential field of study for RTC in bioretention, and the historical detention basins studies have evolved. With this, new hypotheses can be formulated, and multiple objectives can be analyzed in real-time controlled bioretention cells, in addition to assessing the uncertainties of rainfall forecast. The results obtained from the previous studies in bioretention show that there is high potential for future applications, both distributed and centralized RTC. Bioretention cells, as green roofs, also benefit from RTC aiming at water quality improvement for reuse. For green roofs, no specific studies of RTC were found, although there are studies that combine the RTC of the drainage system with passive green roof application scenarios that present good results for reducing peak flow and runoff volume in urban drainage [19,20]. As the RTC studies in NBS evolve, the application of multiple techniques to reach multiple scales of the urban drainage system can be a technological and sustainable alternative to increase urban resilience.

## 5. Conclusions

Challenges and opportunities of applying RTC in NBS were evaluated in this paper. Three techniques commonly applied in urban drainage with different spatial scales were investigated: green roof, bioretention and detention basin. A brief review of their design and results was carried out, both for runoff quantity and water quality. The existing literature on the application of RTC in these techniques was also evaluated, and several opportunities for their application were reported.

For green roofs, only studies using RTC of urban drainage system combined with passive green roofs were found. However, RTC can be also employed directly on green roofs to work similarly with water storage tanks, since the latter already has RTC applications in urban drainage and green roofs have the possibility of storing water. Additionally, green roofs have a pretreatment advantage due to the soil and plants, when compared to water storage tanks.

Bioretention studies have already been carried out with RTC application with positive results. RTC adds versatility to the device, allowing ammonium and nitrate treatment through the variation of aerobic and anaerobic zones. Therefore, it improves the water quality to be reused, such as in households or agriculture. However, there are still scientific gaps for RTC implementation in bioretention, such as multiobjective analyses, centralized controls with the drainage system, in addition to a quantitative analysis of the effects of RTC on bioretention cells.

Studies have also assessed RTC in detention basins. RTC improved the performance of detention basins both for water quantity and quality. In this case, the study opportunities of RTC for the active control of the detention basin involve different designs, since offline detention basins have great potential for improvement with RTC, or multiobjective algorithms for a simultaneous assessment of runoff quantity and quality.

The main difficulties are at the planning stage of the RTC system. These include the decision for the system to be centralized or not, the mathematical models involved in the process, the forecasting system and the costs for monitoring and transmitting data.

The application of RTC in NBS studies on urban drainage are very recent, despite the good results reported in the literature. There are several opportunities that can be explored to optimize the performance of devices, which have already been proven to function satisfactorily. This approach can be used to reduce the problems caused by urbanization and climate change in a more sustainable manner.

**Author Contributions:** Conceptualization, J.B. and E.M. Mendiondo; methodology, J.B. and M.M.; Writing—Original draft preparation, J.B.; Writing—Review and editing, M.M., C.L., M.J. and T.O. (Tassiana Oliveira), T.O. (Thalita Oliveira); visualization, T.O. (Thalita Oliveira); supervision, E.M. All authors have read and agreed to the published version of the manuscript.

**Funding:** This study was funded by FAPESP, grant no. 2014/50848-9 INCT-II (Climate Change, Water Security), CNPq grant no. PQ 312056/2016-8 (EESC-USP/CEMADEN/MCTIC) and FAPESP grant no. 2017/15614-5 "Decentralized Urban Runoff Recycling Facility addressing the security of the Water-Energy-Food Nexus".

**Acknowledgments:** The authors want to thank CNPq, CAPES and FAPESP for scholarships and for research funding.

**Conflicts of Interest:** The authors declare no conflict of interest.

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
