# Peer review of "Nature-Based Solutions and Real-Time Control: Challenges and Opportunities"

_water, doi:10.3390/w13050651_

Round 1

Reviewer 1 Report

The manuscript "Nature-based solutions and Real-time control: challenges and opportunities" is an interesting review paper that discuss about the possibility to increase the NBS (LID in general) efficiency combining with RTC strategies. The paper, in line with the aims of the journal, needs to be improved before to be accepted for publication. Some minor suggestions are below discussed.

Based on introduction section, the authors considers the LID solution and RTC strategy first alone and then they discuss about the possibility to considered them togheter. Moreover, they state that "there are only few studies on the application of the two techniques together". In this regards, this reviewer would sugget to the authors some intersting studies, not yet considered, that take into account the RTC strategy applied to urban drainage system and other ones that considers the integration of LID-NBS solutions with RTC strategies:

  • Piro P., et al., (2019) A Comprehensive Approach to Stormwater Management Problems in the Next Generation Drainage Networks. In: Cicirelli F., Guerrieri A., Mastroianni C., Spezzano G., Vinci A. (eds) The Internet of Things for Smart Urban Ecosystems. Internet of Things (Technology, Communications and Computing). Springer, Cham. https://doi.org/10.1007/978-3-319-96550-5_12
  • Oberascher M., et al. (2019) Smart Rain Barrels: Advanced LID Management Through Measurement and Control. In: Mannina G. (eds) New Trends in Urban Drainage Modelling. UDM 2018. Green Energy and Technology. Springer, Cham. https://doi.org/10.1007/978-3-319-99867-1_134
  • Altobelli, M., et al. (2020). Combined Application of Real-Time Control and Green Technologies to Urban Drainage Systems. Water12(12), 3432. https://doi.org/10.3390/w12123432
  • Maiolo, M., et al. (2020). On the Use of a Real-Time Control Approach for Urban Stormwater Management. Water12(10), 2842. https://doi.org/10.3390/w12102842
  • Xu, W. D., Fletcher, T. D., Burns, M. J., & Cherqui, F. (2020). Real Time Control of Rainwater Harvesting Systems: The Benefits of Increasing Rainfall Forecast Window. Water Resources Research56(9), e2020WR027856. https://doi.org/10.1029/2020WR027856
  • Xu, W. D., et al (2020). Enhancing stormwater control measures using real-time control technology: a review. Urban Water Journal, 1-14. https://doi.org/10.1080/1573062X.2020.1857797

Moreover to increase the quality of the manuscript, this reviewer suggests to add some graphs or table to summarize the studies considered in the manuscript in terms of main objectives and results.  

Author Response

The manuscript "Nature-based solutions and Real-time control: challenges and opportunities" is an interesting review paper that discuss about the possibility to increase the NBS (LID in general) efficiency combining with RTC strategies. The paper, in line with the aims of the journal, needs to be improved before to be accepted for publication. Some minor suggestions are below discussed.

Response: The authors appreciate the comments of the reviewer very much, and we would like to thank the reviewer for the time spent in our manuscript. We have incorporated all the suggestions in the new version of the text. We are glad to have the opportunity to contribute in developing RTC strategies to NBS and urban drainage.

Based on introduction section, the authors considers the LID solution and RTC strategy first alone and then they discuss about the possibility to considered them togheter. Moreover, they state that "there are only few studies on the application of the two techniques together". In this regards, this reviewer would sugget to the authors some intersting studies, not yet considered, that take into account the RTC strategy applied to urban drainage system and other ones that considers the integration of LID-NBS solutions with RTC strategies:

  • Piro P., et al., (2019) A Comprehensive Approach to Stormwater Management Problems in the Next Generation Drainage Networks. In: Cicirelli F., Guerrieri A., Mastroianni C., Spezzano G., Vinci A. (eds) The Internet of Things for Smart Urban Ecosystems. Internet of Things (Technology, Communications and Computing). Springer, Cham. https://doi.org/10.1007/978-3-319-96550-5_12
  • Oberascher M., et al. (2019) Smart Rain Barrels: Advanced LID Management Through Measurement and Control. In: Mannina G. (eds) New Trends in Urban Drainage Modelling. UDM 2018. Green Energy and Technology. Springer, Cham. https://doi.org/10.1007/978-3-319-99867-1_134
  • Altobelli, M., et al. (2020). Combined Application of Real-Time Control and Green Technologies to Urban Drainage Systems. Water12(12), 3432. https://doi.org/10.3390/w12123432
  • Maiolo, M., et al. (2020). On the Use of a Real-Time Control Approach for Urban Stormwater Management. Water12(10), 2842. https://doi.org/10.3390/w12102842
  • Xu, W. D., Fletcher, T. D., Burns, M. J., & Cherqui, F. (2020). Real Time Control of Rainwater Harvesting Systems: The Benefits of Increasing Rainfall Forecast Window. Water Resources Research56(9), e2020WR027856. https://doi.org/10.1029/2020WR027856
  • Xu, W. D., et al (2020). Enhancing stormwater control measures using real-time control technology: a review. Urban Water Journal, 1-14. https://doi.org/10.1080/1573062X.2020.1857797

Response: The authors appreciate the comment of the reviewer and all the papers suggested to incorporate in the literature review. All the papers suggest are of high quality and helped to improve the overview about how RTC can be integrated with NBS in different application scales. In the introduction section we have modified the text to include a discussion about the existence of relevant studies about the joint use of NBS with RTC for different purposes and application scales, working as a passive control, and highlighting the opportunity of new studies that approach the active control (line 75 – 82)

Moreover to increase the quality of the manuscript, this reviewer suggests to add some graphs or table to summarize the studies considered in the manuscript in terms of main objectives and results.  

Response: The authors agree with the reviewer and are grateful for the suggestion. We added Table 3 summarizing the papers that studied RTC and NBS systems, pointing to the study approach, the parameters evaluated (through the column performance criteria) and the type of forecast used in the study. We hope that this table complies with the reviewer's suggestion.

Reviewer 2 Report

The submitted manuscript constitutes essentially a revision paper. Although addressing a very pertinent and up-to-date topic, and promoting an evaluation of the proposed techniques, it is supported by a revision of references. This is not a negative aspect, mainly because of the significant effort to integrate in this analysis/revision valuable contributions and references.

In fact, it presents a very good theoretical framework, based on world class references and complemented with several updated references.

Consequently, it seems to me that this work presents high quality.

Moreover, the text is clear and properly drafted and it is very well structured.

The figures seem to be appropriate.

Author Response

The submitted manuscript constitutes essentially a revision paper. Although addressing a very pertinent and up-to-date topic, and promoting an evaluation of the proposed techniques, it is supported by a revision of references. This is not a negative aspect, mainly because of the significant effort to integrate in this analysis/revision valuable contributions and references.

In fact, it presents a very good theoretical framework, based on world class references and complemented with several updated references.

Consequently, it seems to me that this work presents high quality.

Moreover, the text is clear and properly drafted and it is very well structured.

The figures seem to be appropriate.

Response: The authors are grateful for the reviewer's comment and for the appreciation of the effort made to combine the existing knowledge in these two areas. We hope that the changes made suggested by the other reviewers will help to further improve the quality of the text.

Reviewer 3 Report

The paper presents an application of challenges and opportunities for the application of Real Time Control to Nature-based solutions. This is an interesting topic that needs good contributions.

 They consider three scales (property, street and neighborhood) and select three different techniques (green roof, bioretention and  detention basin) with the objective to evaluate which are the most adequate RTC techniques that may be applied. 

There is a description of NBS techniques that, in my opinion, should be rewritten because it is too generic and in the present form is not specially related to the objective of the paper. 

The revision of the state of the art of RTC application to NBS is only  a description of the literature review, and needs a more critical approach to be useful to the reader. 

The paper needs also some revision of applied cases of RTC use in real management of NBS, not only the bibliographical approach.

I'm sure that the authors may do this work in a new version of the manuscript in order to have a nice paper. 

Author Response

The paper presents an application of challenges and opportunities for the application of Real Time Control to Nature-based solutions. This is an interesting topic that needs good contributions.

Response: The authors would like to thank the reviewer for their time and comments. All the comments were of great value to improve the text quality.

They consider three scales (property, street and neighborhood) and select three different techniques (green roof, bioretention and  detention basin) with the objective to evaluate which are the most adequate RTC techniques that may be applied. 

There is a description of NBS techniques that, in my opinion, should be rewritten because it is too generic and in the present form is not specially related to the objective of the paper. 

Response: Thank you for the comment. The structure of the topic related to the description of the NBS was revised (Section 2 - Opportunities of RTC in different NBS), reducing the very specific descriptions with respect to each of the application scales, since it did not fall within the scope of this article. We focus on a more direct and more critical approach, highlighting the opportunity of RTCs on each device, the gaps that still need to be filled and suggestions for future studies aiming to respond to the gaps raised. The changes can be seen on lines 95 through 273.

The revision of the state of the art of RTC application to NBS is only a description of the literature review, and needs a more critical approach to be useful to the reader.

Response: The state of the art has been updated, adding new papers with results of NBS + RTC approaches, as suggested by reviewer 1. The new papers have already helped to broaden the vision about opportunities with RTC. In addition, according to the reviewer's suggestion 3, a more critical approach was taken, incorporating in addition to the presentation of the results obtained in each study, the limitations and specificities they presented. For this manuscript to be more useful to readers, Table 3 was incorporated, with a summary of all papers with RTC + NBS, presenting the approach used in each article, the parameters evaluated according to their final purpose and the type of forecast used. In addition, we believe that Tables 1 and 2 are also useful for readers to identify where to apply the greatest monitoring efforts for each type of NBS and which is the best type of control and its respective costs for each application scale. In addition, we incorporate more critical discussions for each of the reviews presented. Ex: lines 180-187. We hope that these changes meet what the reviewer asked for. 

The paper needs also some revision of applied cases of RTC used in real management of NBS, not only the bibliographical approach.

Response: Literature review on the experimental application of NBS with RTC were discussed in this paper, that allows application on property scale (green roofs), street scale (bioretention) and neighborhood/watershed scale (detention basins). 4 full-scale experimental papers for the detention basin were incorporated, but still on an experimental scale (GILPIN; BARRETT, 2014; JACOPIN et al., 2001; MIDDLETON; BARRETT; MALINA, 2006; MULLAPUDI et al., 2018; line 250 and 254). For the bioretention basins, 2 papers were cited on a laboratory scale, which evaluated the feasibility of active control to improve water quality (PERSAUD et al., 2019; SHEN et al., 2020; lines 175 and 195). For the green roof devices, only two papers were found that address the theme of RTC, but without active control of the roofs (ALTOBELLI; CIPOLLA; MAGLIONICO, 2020; PRINCIPATO et al., 2017; lines 76, 132, 139). These papers exemplify the first studies with promising results in the application of RTC with NBS for flood and pollution control related to urban drainage. However, there are still no experimental evaluations of its uses for the real management of urban drainage systems. With this review we aim to highlight the next steps that must be taken to advance the application in the field and on a full-scale, incorporating new technologies in the stormwater management. 

I'm sure that the authors may do this work in a new version of the manuscript in order to have a nice paper. 

Response: Thank you. We endeavor to incorporate all suggestions and hope that this new version will be of good quality for publication.

Round 2

Reviewer 3 Report

the authors have satifactorially answered all my concerns